# Large composite fermion effective mass at filling factor 5/2

M. Petrescu [1], Z. Berkson-Korenberg[1], Sujatha Vijayakrishnan [1], K. W. West[2], L. N. Pfeiffer[2] & G. Gervais [1] ✉

The 5/2 fractional quantum Hall effect in the second Landau level of extremely clean two-dimensional electron gases has attracted much attention due to its topological order predicted to host quasiparticles that obey non-Abelian quantum statistics and could serve as a basis for fault-tolerant quantum computations. While previous works have establish the Fermi liquid (FL) nature of its putative composite fermion (CF) normal phase, little is known regarding its thermodynamics properties and as a result its effective mass is entirely unknown. Here, we report on time-resolved specific heat measurements at filling factor 5/2, and we examine the ratio of specific heat to temperature as a function of temperature. Combining these specific heat data with existing longitudinal thermopower data measuring the entropy in the clean limit we find that, unless a phase transition/crossover gives rise to large specific heat anomaly, both datasets point towards a large effective mass in the FL phase of CFs at 5/2. We estimate the effective-to-bare mass ratio $m^*/m_e$ to be ranging from ~ 2 to 4, which is two to three times larger than previously measured values in the first Landau level.

The concept of an effective mass is ubiquitous in solid-state physics and for years it has been used in semiconductors to understand the transport properties of electrons under the influences of a variety of fields. In the case of clean two-dimensional electron gases (2DEGs) described by Fermi liquid theory, the renormalization of the electron mass into an effective mass $m^*$ due to interactions provides important insights into the dynamics of its elementary excitations called quasiparticles. This effective mass can be linked to several thermodynamic quantities that can be experimentally measured, and it provides important guidance for theory work aimed at understanding the many-body electronic states within a set of conditions such as magnetic fields strength, electron densities, etc. In this regard, much progress was made in the past in the first Landau level (FLL), however the opposite cannot be more true in the second Landau level (SLL) where measurements of $m^*$ are entirely absent. This is the object of this work whereby the effective mass was experimentally estimated in the SLL of a clean 2DEG.

More than three decades since its discovery[1], the 5/2 fractional quantum Hall effect (FQHE) in the SLL remains the source of extensive research fuelled by its predicted non-Abelian topological order described by a many-body Pfaffian state[2,3], anti-Pfaffian[4,5], or its particle–hole symmetric form[6,7]. This two-dimensional many-body quantum state has recently received strong credence from thermal transport measurements in isolated edges[8–10]. At temperatures above the many-body energy gap $\Delta_{5/2}$, a Fermi liquid phase of spin-polarized composite fermions (CFs)[11–13] is believed to occur and the formation of a Fermi sea at $v = 5/2$ has been confirmed by surface acoustic waves[14] and geometric resonance[15] experiments.

In the FLL, where no FQHE occurs at $v = 1/2$ and $v = 3/2$, the CFs Fermi sea picture has also been confirmed previously by surface acoustic wave experiments[16,17]. Several experiments have also been designed to extract the effective mass $m^*$, and these include thermodynamic measurements of 2D electron or hole gas (2DEG/2DHG) probing the thermopower[18,19], or simply the energy gap[20–22] of the FQH states in the vicinity of $v = 1/2$ and $v = 3/2$. In all cases, the effective mass $m^*$ of the CFs was found to be close to the bare electron mass $m_e$, and ranging from ~0.7 to $1.3m_e$. In spite of these advances in the FLL, the situation differs greatly in the SLL, where surprisingly little is known

[1]Department of Physics, McGill University, Montreal, Quebec H3A 2T8, Canada. [2]Department of Electrical Engineering, Princeton University, Princeton, NJ 08544, USA. ✉e-mail: gervais@physics.mcgill.ca

regarding the thermodynamic properties at $\nu = 5/2$ in the Fermi liquid phase, and as a result the effective mass $m^*$ of its CFs remains entirely unknown experimentally. To our knowledge, this manuscript is the first to provide an experimental estimate on the CFs effective mass in the Fermi liquid phase at $\nu = 5/2$.

In the CFs Fermi liquid phase, the specific heat is expected to exhibit a linear temperature dependence given by[21]

$$C_{CF} = \frac{\pi(1+p_{CF})m^* k_B^2 T}{6\hbar^2 ne} = \gamma_{CF} T, \qquad (1)$$

where $p_{CF}$ is the CF impurity scattering parameter, $m^*$ is the effective mass, $k_B$ is the Boltzmann constant, $\hbar$ is the reduced Planck's constant, $n$ is the 2DEG density, $e$ the electron charge and $\gamma_{CF}$ is the CFs specific heat linear constant. This model assumes a parabolic dispersion and also that electrons are maximally polarized. While there is currently a debate as to whether or not the $\nu = 5/2$ FQHE could host a Dirac-like dispersion[6], it is unclear whether this would affect its putative CF Fermi liquid phase. In the case of the second assumption, the full polarization of the electrons has been validated experimentally at $\nu = 5/2$ by resistively detected NMR experiments[23,24]. The same formula applies at all half-integer filling fractions $\nu_i = i + 1/2$, as stated in ref. 21, with the caveat that in the second LL at $\nu = 5/2$, when compared to the lowest LL ($i = 0, 1$), the effective mass $m^*$ could differ. Unless mentioned otherwise, we chose in the analysis below $p_{CF} = 0$ which assumes a weak energy dependence of the scattering rate, as assumed in ref. 19,25.

The size of the Fermi liquid sea $\nu = 5/2$[14,15] has been previously reported to be large, i.e., extending over 0.1 T in magnetic fields near $\nu = 5/2$[14]. As a consequence, it extends to magnetic fields larger than the deviation from the exact filling factor ($\nu^* = 0$) in our work, where $\nu^* \equiv [\nu - \nu_0]$ is the non-exact filling factor with $\nu_0 = 5/2$. Thus, the assumption of a well-formed Fermi liquid phase of CFs for our measurements reported at $\nu^* \neq 0$ should also be valid. Finally, we note that in a Fermi liquid phase, the value of $C/T$ should become a constant, and this has been exemplified spectacularly in the normal Fermi liquid phase of $^3$He even in the presence of disorder[26]. Although the data reported in our work are taken in the FQH condensed phase, we expect that $C/T$ should tend towards a constant ($\gamma_{CF}$) as temperature increases towards the Fermi liquid phase, therefore allowing for the CFs effective mass $m^*$ to be estimated.

## Results and discussion
### Specific heat over temperature ratio
The specific heat data was acquired at discrete magnetic fields at, and in the vicinity of the $\nu = 5/2$ FQH state, as depicted by the green dotted and dashed lines in Fig. 1b. Figure 2 shows the temperature dependence of $C/T$ ratio (blue filled circles) in the range 20 mK $\leq T \leq$ 80 mK. The solid blue is a guide-to-the-eye fit employed to find the limiting value in the CFs Fermi liquid phase. To lend support to our data and gain information on the trend of $C/T$ at temperatures above 80 mK, we show in the same figure the $C/T = dS/dT$ ratio extracted from thermopower[27] for a 2DEG with similar electron density and mobility. In the clean limit, the thermopower $S_{xx}$ extracted from the thermal voltage $\Delta V_{xx}$ developing in the presence of a thermal gradient $\Delta T$ provides a measure of the entropy $S$ per electron charge, i.e., $-S_{xx} = \frac{\Delta V_{xx}}{\Delta T} = \frac{S}{eN_e}$, where $e$ is the electron charge, and $N_e$ the total number of electrons in the 2DEG. The entropy per unit electron measured by thermopower is shown in the inset of Fig. 2 (upper panel) and the solid green line is a fit of these data. The green curve shown in the main panel of Fig. 2 was then obtained by taking the numerical derivative of the entropy, hence providing the $C/T$ ratio. The green symbols in the main panels are markers denoting the temperature at which the thermopower was measured. In the range of temperature overlap between the specific heat and thermopower experiments, we find

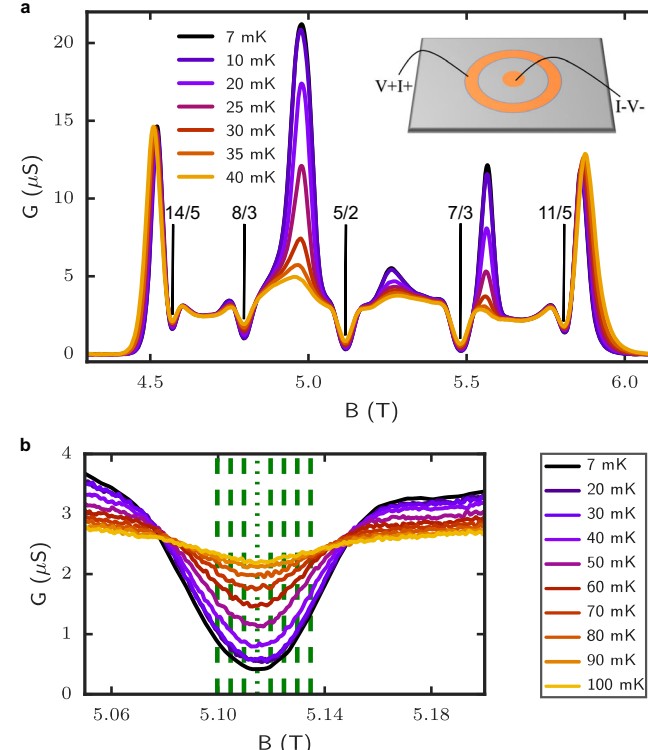

**Fig. 1 | Conductance measurements in the second Landau level. a** Conductance versus magnetic field at different temperatures. The most prominent FQH states are indicated by vertical lines. A cartoon of the Corbino sample is shown in the inset. **b** Zoom-in of the conductance temperature dependence for 5/2 FQH state. The thin green dotted line shows the exact filling factor $\nu = 5/2$ (i.e., $\nu^* = 0$), whereas the dashed lines are non-exact filling factors (i.e., $\nu^* \neq 0$) for which specific heat measurements were taken and where the effective mass could be estimated.

close *quantitative* agreement between the two datasets. This is certainly remarkable given how fundamentally distinctive the two experiments are, both in their experimental details and experimental execution.

### Effective mass estimation from limiting value of *C/T*
We now turn our attention to the limiting value of $C/T$ in the Fermi liquid phase for both datasets that are shown in the top panels of Fig. 2, the dashed blue (from specific heat) and dashed green (from thermopower) lines. These limiting values were estimated from the trend of the data at 200 mK where it is expected that a CFs Fermi liquid phase is formed, with a linear specific heat (and entropy) temperature dependence. This is particularly well exemplified with the thermopower data at $\nu^* = 0$ where the $C/T$ ratio clearly tends towards a constant, as expected in the CF Fermi liquid phase. The limiting value of $C/T$ provides the constant $\gamma_{CF}$ (see Eq. (1)), and in turn allows us to estimate the CF effective mass. With a CF impurity scattering parameter $p_{CF} = 0$, an assumption recently used in thermopower work at $\nu = 1/2$[19], and for which theory works[21] had found it to be small ($p_{CF} = 0.13$), we find a CF effective mass ranging from $m^* \simeq 2.7 m_e$ (with thermopower) to $m^* \simeq 3.9 m_e$ (with specific heat). In any case, it is substantially larger than the bare electron mass and this is emphasized in the upper panel of Fig. 2 by a red dashed line showing the CF Fermi liquid limiting value if $m^* = m_e$.

### Effective mass estimation from *C/T* and entropy considerations
We also considered another approach to estimate the CF effective mass, making use of the area under $C/T$ curve which is a measure of the the entropy, $S(T) = \int_0^T \frac{C}{T'} dT'$. The area up to 200 mK is shown for both datasets in the lower panel of Fig. 2 with the shaded blue region

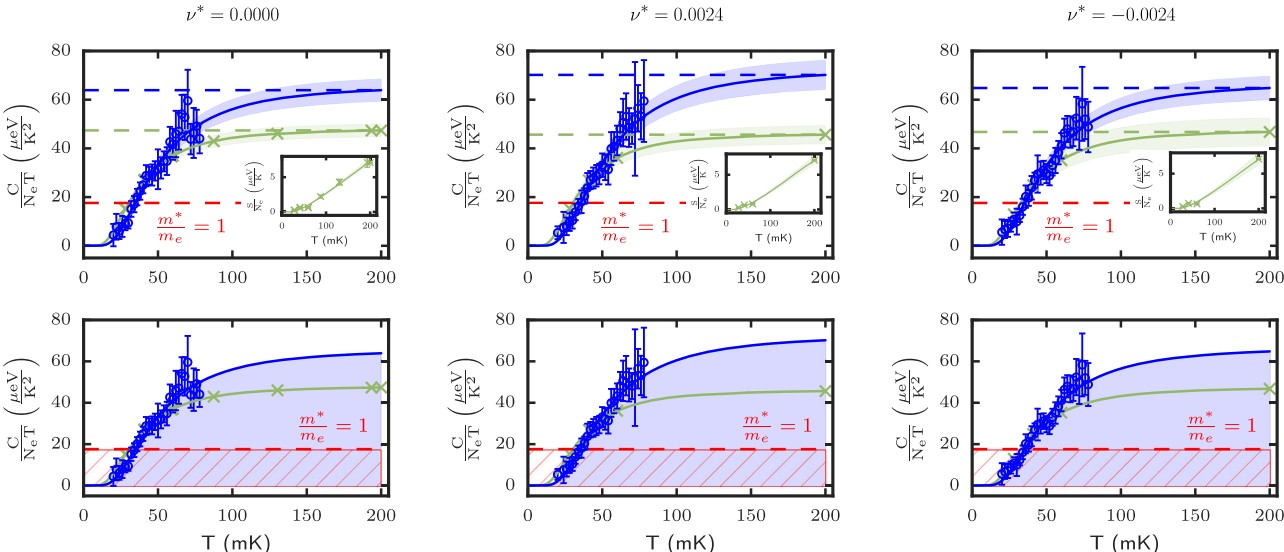

**Fig. 2 | Extraction of composite fermion effective mass from $C/T$ and entropy.** Temperature dependence of $C/T$ at $\nu = 5/2$ with corresponding relative filling factors $\nu^* = 0$ and $\nu^* = \pm 0.0024$. Upper panels: specific heat data is shown with blue circles and the blue curve is a guide-to-the-eye. The blue-shaded region is an uncertainty bound. The dashed blue line denotes the limiting value of $C/T$ in the Fermi liquid phase. The solid green line is the $C/T$ extracted from thermopower data shown in the inset by green crosses[27]. Its limiting value in the Fermi liquid phase is shown by the green dashed line. From the limiting values denoted by the green and blue dashed lines, composite fermion mass was estimated (see main text). The red dashed line shows the Fermi liquid limiting value if the effective mass $m^* = m_e$. Lower panels: same data as the upper panels but with the blue-shaded area depicting the total entropy from 0 to 200 mK from which the effective mass can also be estimated (see main text). The red-hatched region depicts graphically the total entropy from 0 to 200 mK if the effective mass $m^* = m_e$. The error bars for the specific heat are showing the statistical errors propagated from the measurements of $\tau$ and $K$, see the Supplementary Information (SI) section 9. The error bars in thermopower data denote the uncertainty extracted from the local noise.

corresponding to the specific heat data, and the hatched red region emphasizing the area if the CF effective mass $m^* = m_e$. For clarity, we only provide the $C/T$ thermopower data (green markers), understanding that its entropy as defined by the area under the green curve will be considered in the analysis below. Performing a simple visual inspection of the blue-shaded area and the red-hatched region where $m^* = m_e$, it is clear that the CF effective mass at $\nu = 5/2$ must be significantly larger than the bare electron mass.

The third law of thermodynamics and conservation of entropy dictates that the entropy in the condensed FQH phase must recover the Fermi liquid value entropy at higher temperatures, i.e $\int_0^T (\gamma_{CF} - \frac{C}{T}) dT' = 0$, with $T'$ in the CF Fermi liquid phase. As an example, specific heat measurements in $^3$He at ultra-low temperatures have spectacularly shown the $C/T$ area in its superfluid phases to recover the Fermi liquid value at the critical BCS transition temperature $T_c$, even with the presence of a BCS-specific heat peak discontinuity as well as engineered weak disorder (see ref. 26 and references therein). Building on this and the constraints given by the third law of thermodynamics, integration of the area to 200 mK temperature which leads to the entropy at that temperature, we find the CF effective mass ranging from $m^* \simeq 2.1 m_e$ (with thermopower) to $m^* \simeq 2.8 m_e$ (with specific heat). While at first sight there may appear to be a discrepancy with the CF effective mass estimated from the limiting value of $C/T$, this is due to the missed entropy stemming from $C/T$ not being entirely constant at 200 mK. This being said, we stress that there is simply too much area under the curve of $C/T$ versus $T$ for the CF effective mass to be equal to the bare electron mass (red-hatched area). This is true assuming a scattering parameter $p_{CF}$ that is similar to that calculated in the first Landau level at half fillings (see discussion below).

## Considerations in the event of a specific heat anomaly and thermodynamic transition

Here, we consider the case where there could be a specific heat anomaly due to a thermodynamic phase transition. Although such a

transition is a priori not expected between a CF Fermi liquid and the FQH ground state, we consider it because if it were to occur, the temperature evolution of the $C/T$ data could perhaps not follow the guide-to-the-eye trend shown in Fig. 2. Thus, if such a phase transition were to occur, it could in principle lower the total area under the curve of $C/T$ versus $T$, as is known for example in superconductors and superfluid $^3$He[26], and hence lower the $m^*/m_e$ ratio closer to one. We therefore hypothetically consider at which temperature $T_c$ a specific anomaly could occur so that the constraint $\int_0^{T_c}(\gamma_{CF} - \frac{C}{T})dT' = 0$ is respected for a CF effective mass equal to the bare electron mass, $m^* = m_e$. To illustrate it, we show in Fig. 3 by a red shade the $C/T$ ratio *above* the CF Fermi liquid value (with $m^*/m_e = 1$, red dashed line) which must compensate the area below the Fermi liquid value at low temperatures, also shown by a red shade. In this figure, the thick vertical dashed line denotes the temperature ($T_c \simeq 60$ mK) at which this hypothesized specific heat anomaly should occur for the entropy to be conserved with the condition $m^* = m_e$. This temperature is well within the observable range of the specific heat experiment, and we do not observe a clear specific heat anomaly whose decrease would have had to reach the red dashed line in order for $m_{CF} = m_e$. Moreover, if hypothetically such anomaly would occur at temperatures above 80 mK, entropy conservation would enforce the Fermi liquid CF effective mass to become larger than the $m_e$, even if such transition were to be above 200 mK temperature.

We also considered the case where a hypothetical thermodynamic transition would be first order, hence with an entropy discontinuity (latent heat). In this case, the change in slope of the entropy with temperature would lead to specific heat discontinuity which would have been missed in both the specific heat and thermopower experiments up to 200 mK. We therefore conclude that such thermodynamic transition is unlikely, and summing up all the aforementioned arguments, we conclude that the CF effective mass at $\nu = 5/2$ must be large due to: (i) the $C/T$ versus $T$ trends tending towards a limiting value in both experiments; (ii) the area calculated under the curve (entropy) that is much larger than for a bare electron mass; (iii) constraints from

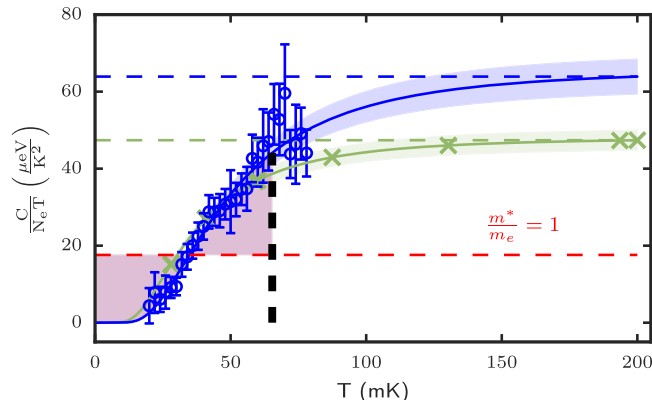

**Fig. 3 | Entropy considerations in the event of a specific heat peak anomaly.** The $C/T$ versus $T$ data are shown at $v = 5/2$ (exact filling) with the same symbol convention as in the upper panel of Fig. 2, except for the red shaded area denoting the area above and below the CF Fermi liquid value if $m^* = m_e$. The vertical dashed line marks the temperature at which a sharp decrease in $C/T$ should have occurred for the entropy to recover the CF Fermi liquid value for $m^* = m_e$. The data show no evidence for a large specific heat peak decrease or jump, and should it occur at higher temperatures, constraints from the third law of thermodynamics would imply that the CF effective mass $m^*$ is larger than the bare electron mass, $m_e$. The data and error bars of $C/T$ are the same as in Fig. 2.

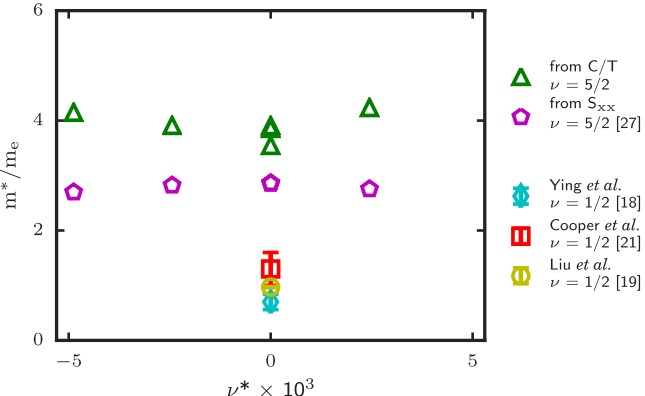

**Fig. 4 | Composite Fermion effective mass at and close to 5/2 filling factor.** Our estimated effective mass values (in units of bare electron mass) obtained from the limiting value of $C/T$ are shown versus $v^*$, and are compared with previous measured values in the first Landau level. At exact filling fraction $v = 5/2$, three values are reported since the effective mass $m^*$ was estimated from three distinct specific heat datasets. The original data (cyan diamond) from ref. 18 is shown with $p_{CF} = 1$ together with $p_{CF} = 0.13$ upon the analysis of Cooper et al.[21] (red square). In the most recent thermopower work at $v = 1/2$ by Liu et al.[19], $p_{CF} = 0$ was used (yellow circle). The error bars of the first Landau level are the previously reported values results, see refs. 18,19,21.

the third law and area consideration make it unlikely that a thermodynamic transition could occur and lower the effective mass value; and (iv) the CF effective mass is most likely large, and bounded in the range of $m^* \sim 2 - 4 m_e$.

## Comparison with the first Landau level

In Fig. 4, the $m^*/m_e$ ratio is shown for all filing factors investigated in this work, i.e. at exact filling factor ($v^* = 0$) and in in the $v = 5/2$ vicinity ($v^* \neq 0$). In the absence of effective mass measurements in the Fermi liquid phase of CFs at 5/2, we compare our results with previous experiments in the lowest LL at $v = 1/2$. While there are considerable differences between previously reported values at and near 1/2 filling factor[18,19], the reported values place the CF effective mass in the range of 0.7 to $1.3 m_e$, i.e. relatively close to the bare electron mass, and hence significantly lower than at $v = 5/2$. It is worth noting that choosing $p_{CF} = 1$, would reduce our reported value by roughly a half, but it would still be three times larger than the quoted values in the FLL using the same scattering parameter value. However, we cannot rule out the possibility for $p_{CF}$ to be close to unity in the second LL which, if it were the case, would bring the CF effective masses closer in values. New theory work is certainly required here to clarify the exact role played by $p_{CF}$ in the second Landau level.

Other previous works also focused on energy gap measurements and Dingle analysis to estimate the CF effective mass value near half-filling. Some experiments suggest strong dependence on the effective field $B^* = B - B_v$ diverging as $v \to 1/2$[20]. Similar conclusions were drawn from energy gap measurements near $v = 1/4$ with $m^* \approx m_e$[28] and in the vicinity of $v = 3/4$[29]. However, a recent experiment probing the effective mass $m^*$ near $v = 1/2$[22] did not show any clear divergence as $v \to 1/2$ (from both sides) within their data resolution and uncertainty. Rather, the effective mass $m^*$ near $v = 1/2$ was fitted and a magnetic field dependence proportional to $\sqrt{B}$ was deduced, with an effective mass around $v = 1/2$ roughly five times smaller ($m^* \approx 0.6 - 0.7 m_e$) than our reported value at $v = 5/2$.

Finally, we note that the effective mass ratio shown in Fig. 4 does not exhibit a clear dependence on the filling factor $v^*$, and hence with a small magnetic field $B$ deviation from $v = 5/2$. This may not be surprising given the very small magnetic field deviation used here, i.e., $\Delta B \in [-0.005, +0.01] \, T$ over the entire filling factor $v^*$ axis of Fig. 4. This

is much smaller than the expected extent of the CF Fermi liquid sea (up to 0.1 T) that was found in surface acoustic waves experiments[14,15]. Thus, the small magnetic field deviation used in our work would account for a minuscule effective mass difference of $\Delta m^* \sim 0.01 m_e$ if it were to follow a $\sqrt{B}$ dependence, one that cannot be resolved by our experiment. All considered the flat trend of the CF Fermi liquid effective mass deduced near 5/2, and shown in Fig. 4, may not be surprising.

To summarize, by probing the specific heat of the bulk 2DEG at both exact and non-exact fillings near $v = 5/2$, and examining the limiting behavior of $C/T$ with temperature as well as from integration of the $C/T$ ratio, a very large effective mass was found and estimated to be two to four times the bare electron mass. While it differs much from previously measured effective masses in the first Landau level, it may not be surprising given that at a 5/2 filling fraction a FQH state can form due to residual electron–electron interaction and a CF Fermion sea instability. To our knowledge, this is the first report of the effective mass at that filling factor, and we hope that our findings will help further understand exactly *how* the enigmatic 5/2 FQH state can form in the second Landau level at half-integer filling.

## Methods

All data presented in this manuscript was taken in a 2DEG formed in the Corbino geometry, which ensures that only the bulk is probed, i.e., it excludes any edge contributions. The wafer is a GaAs/AlGaAs heterostructure with quantum well width 30 nm, an electron density $n_e = 3.08 \pm 0.01 \times 10^{11}$ cm$^{-2}$, a raw wafer 2DEG mobility of $25 \times 10^6$ cm$^2$/V·s and a measured Corbino sample mobility of $22 \pm 2 \times 10^6$ cm$^2$/V·s using the procedure outlined in refs. 30–32. The Corbino sample has a central contact with an outer radius $r_1 = 0.25$ mm and an inner ring contact with radius $r_2 = 1.0$ mm. The contacts were first patterned using UV lithography and then fabricated by e-beam deposition of Ge/Ni/Au/Au layers with corresponding 26/54/14/100 nm thickness. In the last step of the fabrication, the contacts were annealed in the presence of $H_2N_2$ at 420 °C for 80 s. A cartoon representation of the Corbino sample is shown in the inset of Fig. 1a. The sample was illuminated by a red LED during cool-down until temperature reached 6 K to enhance the 2DEG density and mobility.

The magneto-conductance measurements near $v = 5/2$ are shown in Fig. 1a, b. A standard lock-in sampling technique was used with a bias

of 0.5 mV and a frequency of 13.3 Hz. All specific heat data was acquired at fixed magnetic fields, with square wave biases ranging from 0.5 to 5.0 mV at 10.5 kHz. The circuits for all measurements are shown in the SI. The high-frequency signals were measured with a Zurich HF2LI lock-in digitizer. Averaging over a million samples of pulse trains and signals was necessary in order to improve the signal-to-noise ratio and to reduce the overall uncertainty. Furthermore, the parasitic wire resonance peaks were eliminated using the shift and subtract method presented in SI. Here, temperature dependence of conductance was used as a thermometer in order to determine electron temperature as a function of applied power, allowing for the extraction of thermal conductance $K$. Due to the non-linear temperature dependence of the applied power, a minimum amount of four data points was kept for the linear fitting procedure of the thermal conductance $K$ outlined in SI. The thermal relation time $\tau$ is extracted using an exponential decay fit of the 2DEG response measured by the digitizer (more details in SI). Heat capacity is simply given by $C = K\tau$ and the specific heat per electron by $c = C/N_e k_B$, where $N_e$ is the number of electrons in the Corbino annular ring.

The thermopower entropy data were determined using a digitized version of the data of ref. 27, with the uncertainty determined by visual inspection of the local noise, and overall background fluctuation. A guide-to-the-eye fit (as showcased in the inset of Fig. 2, top panel) was used to determine the slope at specific temperatures corresponding to thermopower entropy data. This slope corresponds to the ratio of specific heat and temperature which is presented in Figs. 2 and 3.

## Data availability
The data presented in this work are available on the McGill University Borealis Dataverse Project under the collection Gervais Lab Physics Data.

## Code availability
All the code as well as the routines used for the data acquisition and analysis of this work are available from the corresponding author upon request.

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

## Acknowledgements

This work has been supported by NSERC (Canada), FRQNT-funded strategic clusters INTRIQ (Québec) and Montreal-based CXC. The work at Princeton University is funded in part by the Gordon and Betty Moore Foundation's EPiQS Initiative, Grant GBMF9615 to L.N. Pfeiffer, and by the National Science Foundation MRSEC grant DMR 2011750 to Princeton University. Sample fabrication was carried out at the McGill Nanotools Microfabrication facility. We would like to thank B.A. Schmidt and K. Bennaceur for their technical expertise during the fabrication and the earlier characterization of the Corbino sample, F. Boivin for assistance during the preparation of the revised manuscript, and R. Talbot, R. Gagnon, and J. Smeros for technical assistance.

## Author contributions

M.P. and G.G. conceived the experiment. K.W.W. and L.N.P. performed the semiconductor growth by molecular beam epitaxy and provided the material. M.P. performed the electronic transport measurement in the Corbino samples at low temperatures, with the assistance and expertise of S.V. M.P. performed the data analysis, helped by Z.B.K. for the development of computer routine. M.P. and G.G. wrote the manuscript, and all authors commented on it.

## Competing interests

The authors declare no competing interests.
