## [Peer Review File · Nature Communications]

REVIEWER COMMENTS

Reviewer #1 (Remarks to the Author):

In the manuscript entitled “Large composite fermion effective mass at filling factor $5/2$ ”, the authors measured the specific heat at/near $\nu=5/2$ in the fractional quantum Hall state at low temperatures and estimated the trend of the specific heat over higher temperatures in the composite fermion regime. Based on the experimental results and theoretical predictions, the authors estimated the effective mass of the composite fermions at $\nu=5/2$ to be much larger than the original mass. The authors also analyzed thermopower data from a reference work and reached a consistent conclusion about the large effective mass. The effective mass of the composite fermions in the first Landau level has been measured through a few different methods in the past. However, no findings in the second Landau level, the most important one and the most challenging one, were reported before. In addition, reliable specific heat measurement for a real two-dimensional system is anticipated for a long time. The larger value at $\nu=5/2$ and the difference between $\nu=5/2$ and $\nu=1/2$ will definitely be of interest to researchers in the fractional quantum Hall community. We (this report is a joint review) appreciate the efforts for the challenging heat capacity measurement. However, we do have a few suggestions and concerns regarding the measurement and the explanation.

1. The data are based on the heat capacity measurement. The measurement setup and the uncertainty should be described in the main text and emphasized in the Supplementary Information for readers. For example, for the two-dimensional electron gas confined in semiconductor, how much heat capacity from the phonons will be measured as well, given the T^5 coupling between electron and phonon. Probably that is also the reason that heat capacity measurement didn't extend to higher temperatures for the authors. Besides electron-phonon coupling, how did the authors partial out the contribution from the ohmic contacts and leads? The T linear contribution from metals could be a significant uncertainty source at low temperature limit. In addition to the estimate, is there any control experiment the authors can provide?

2. We are deeply concerned about the entropy analysis.

First, at ultra-low temperatures below 100 mK, any heat capacity measurement unavoidably includes the background, and the unreliability of the thermometer also causes the deviation from the “real” data. Integral from heat capacity to get entropy is controversial because all the errors from heat capacity will accumulate. Heat capacity is more difficult to measure at lower temperatures, and C/T gives more weight for the data with larger uncertainty. We think the authors should provide further evidence in this regard, especially the quantitative evidence.

Second, getting the entropy from the specific heat is dangerous for another reason. In any realistic measurement, people can measure heat capacity to only a lowest limit (let's say ~ 20 mK in this

manuscript). One can only calculate the entropy from 20 mK but there is no apparent way to guarantee C/T below 20 mK is unimportant, which change the shape of the entropy curve after integral.

3. Minor issues:

The authors measured the specific heat between 20mK and 80mK and extended the trend to 200 mK. The trend, i.e. the guide-to-the-eye curve, is very critical in the estimation of the effective mass. We recommend that the authors describe in more details how they decided the trend.

Can the authors explain why the impurity scattering parameter p_{CF} is 0 in their sample? As the authors stated in the main text, $p_{CF}=1$ would reduce the value by a half, which could make the effect mass value at $\nu=5/2$ much closer to the original value.

Reviewer #2 (Remarks to the Author):

The manuscript from M. Petrescu et al. reports on the specific heat measurement of the effective mass of composite fermions (CF) in the composite Fermi liquid at the filling factor $\nu=5/2$. The result is about two to four times the bare electron mass, which is significantly larger than the corresponding value in the half-filled lowest Landau level (LLL).

In my opinion, the experimental procedures and analysis have been clearly explained in the manuscript. The studies is also crucial in the understanding of CF theory by highlighting the possible difference between CF in the half-filled LLL and half-filled higher LLs. Therefore, I would recommend a publication of the manuscript in Nature Communications if the authors can address the following comments and suggestions:

1. The effective mass was actually estimated by extrapolating the results in the $\nu=5/2$ FQH state to the high-temperature regime in which the composite Fermi liquid is believed to form. While this assumption seems to be reasonable and further justified by the thermopower data, is there any more direct evidence for this claim? For example, any estimate of the gap $\Delta_{5/2}$ for the $\nu=5/2$ FQH state in the sample? Is the temperature 200 mK large compared to $\Delta_{5/2}$?

2. I think the C/T curves in Figs. 2 and 3 for $\nu^* = 0$ show a little peak at $T \approx 70$ mK and drop at a higher temperature. This trend looks similar to what happens in the BCS transition. It is hard to rule out this possibility without more data points in higher temperature (beyond 80 mK). Also, the temperature of the apparent peak is close to the transition temperature $T_{c'} \approx 60$ mK predicted from the third law of thermodynamics argument in the manuscript. Can the authors clarify or comment on that?

3. In the first paragraph, it is mentioned that the $5/2$ FQHE may be described by a Pfaffian state or its particle-hole symmetric form. Meanwhile, recent numerical results suggested that the $5/2$ FQH state may be described by the anti-Pfaffian state (particle-hole conjugate of the Pfaffian state). Since the nature of this FQH state is still under debate, it is better to mention and include the references on the anti-Pfaffian state as well.

4. To the best of my knowledge, the derivation of Eq. (1) in the original work by Cooper, Halperin, and Ruzin assumed that the composite fermions have a parabolic dispersion. There is a recent debate on whether the composite fermions are actually Dirac-like or not. Does it matter in the present work? I suggest the authors to clarify the assumption in Eq. (1).

Reviewer #3 (Remarks to the Author):

Reviewer #4 (Remarks to the Author):

Under a high perpendicular magnetic field at very low temperatures, a two-dimensional electron system may enter the fractional quantum Hall state at the Landau level filling factor $\nu = 5/2$. Quasiparticles at the $5/2$ state may obey non-Abelian statistics, and thus they have been intensively studied due to the fundamental interest in strongly correlated electron systems and the possible applications toward topological quantum computation. This manuscript reports on the properties of the electron system near $\nu = 5/2$ at high temperatures, where the system does not enter the $5/2$ -fractional quantum Hall state. It is known that, at such high temperatures, the electron system is in a Fermi liquid phase of spin-polarized composite fermions. The authors of the present manuscript

claim that, while the effective mass of the composite fermions in the Fermi liquid phase near $\nu = 1/2$ and $3/2$ has been reported in the previous papers, the present manuscript report is the first to provide the experimental estimation of the effective mass near $\nu = 5/2$.

I do not recommend publishing this manuscript in Nature Communications. The reasons are as follows.

1. This manuscript reports a large effective mass near $\nu = 5/2$ but fails to explain why this is an important finding. Even if the authors are correct in claiming that this is the first observation of the effective mass at $\nu = 5/2$, I cannot accept the result as impactful enough for general readers of Nature Communications without sufficient evidence that the observation is scientifically important.

2. There needs to be an explanation of the mechanism why the effective mass is greater at $\nu=5/2$.

3. More discussion needs to be provided for the relationship between the fact that the effective mass is large and that the $\nu=5/2$ non-Abelian fractional quantum Hall effect occurs at lower temperatures.

4. Finally, while the data provided in this manuscript may be sufficient to demonstrate the large effective masses qualitatively, it lacks clarity in representing the difference between the first and second Landau levels and precise quantification.

It is premature to immediately judge that this paper interests the general readers because of its flavor of the $5/2$ state in fashion. I respect the data presented here, but that is from a specialized perspective on quantum Hall systems. In summary, this paper should be published in a more specialized journal.

Below I comment on some of my concerns.

5. The authors emphasize the value of the effective mass when assuming $P=0$, which was not fair.

6. Determining the effective mass based on the guide-to-the-eye lines (blue lines in Fig. 2, for example) seems non-scientific. Isn't there a better way?

7. The insets in Fig. 2 are too small to recognize what is inside.

8. In the first paragraph on page 3, the authors emphasize the reliability of this study by claiming that the results of the two methods are in quantitative agreement at relatively low temperatures. However, the data in the high-temperature region is important for effective mass estimates, where the two methods do not agree. Therefore, the authors' claim is not fair.

9. The authors claim in the caption of Figure 3 that there are no specific heat peaks. However, it is not impossible to see a peak in the data. I observe that some data show the peak-like structure, and others do not. The claim sounds biased.

10. The authors compare their data near $5/2$ with data near $3/2$ and $1/2$ in the previous literature, but for an accurate comparison, the authors should obtain data at $3/2$ and $1/2$ by themselves and compare them with the data at $5/2$.

Reviewer #1 (Remarks to the Author):

In the manuscript entitled “Large composite fermion effective mass at filling factor $5/2$ ”, the authors measured the specific heat at/near $\nu=5/2$ in the fractional quantum Hall state at low temperatures and estimated the trend of the specific heat over higher temperatures in the composite fermion regime. Based on the experimental results and theoretical predictions, the authors estimated the effective mass of the composite fermions at $\nu=5/2$ to be much larger than the original mass. The authors also analyzed thermopower data from a reference work and reached a consistent conclusion about the large effective mass. The effective mass of the composite fermions in the first Landau level has been measured through a few different methods in the past. However, no findings in the second Landau level, the most important one and the most challenging one, were reported before. In addition, reliable specific heat measurement for a real two-dimensional system is anticipated for a long time. The larger value at $\nu=5/2$ and the difference between $\nu=5/2$ and $\nu=1/2$ will definitely be of interest to researchers in the fractional quantum Hall community. We (this report is a joint review) appreciate the efforts for the challenging heat capacity measurement. However, we do have a few suggestions and concerns regarding the measurement and the explanation.

Author’s response: We thank both referee 1 and 3 for their joint and thorough review of our manuscript, for their positive comments, and for their appreciation regarding our advance in measuring the specific heat of a 2DEG at very low temperatures.

1. The data are based on the heat capacity measurement. The measurement setup and the uncertainty should be described in the main text and emphasized in the Supplementary Information for readers. For example, for the two-dimensional electron gas confined in semiconductor, how much heat capacity from the phonons will be measured as well, given the T^5 coupling between electron and phonon. Probably that is also the reason that heat capacity measurement didn’t extend to higher temperatures for the authors. Besides electron-phonon coupling, how did the authors partial out the contribution from the ohmic contacts and leads? The T linear contribution from metals could be a significant uncertainty source at low temperature limit. In addition to the estimate, is there any control experiment the authors can provide?

Author’s response: We thank the reviewers for their comment. Here is the answer to the reviewers’ first concern:

- (a) While we acknowledge that a description of the measurement setup and the uncertainty analysis in the main text would be useful for the reader’s understanding, the reason we have excluded it from the main text is because most of it had already been published in [1] and detailed in a thesis [2] available on the GervaisLab web page. The discussion of the error propagation and the measurement circuit that was used were already present in the supplementary material (SM). This being said, we have greatly enlarged the SM so that the overall manuscript is now more ‘standalone’.
- (b) We thank the reviewers for asking us about the role played by the phonons, as one would *a priori* expect them to play a role via electron-phonon interactions. However, our

experiment was specifically designed to exclude the phonon contribution by leveraging the time scale separation of electron and phonon responding to a heat pulse. Our scheme is based on a previous *tour de force* experiment [3] at higher temperatures, albeit in our case joule heating is applied *in situ* the 2DEG. The important point is that there is a separation of time scale where the 2DEG thermal relaxation time is a few microseconds whereas the phonons have a tens of milliseconds response. Therefore, by measuring “fast” we can access the thermal relaxation time of the 2DEG directly. This has been discussed at length in our previous reference [2].

- (c) Regarding the contact, this is a subtle and important point. Perhaps the most important aspect is that our experiment was conducted in the Corbino geometry. Since there is no edge in this 2D geometry, the conductance drops near to “zero” in a QH or FQH state. For example, at filling factor $\nu=5/2$, the resistance of the 2DEG in our Corbino is at least 500 k Ω , whereas our contact resistance is on the order of a few Ω only (note this is somewhat remarkable and is due to the Corbino geometry having very large contact area). The thermal relaxation in the Fermi liquid contact reservoir is expected to be much faster than in the 2DEG (nanoseconds), and entirely unresolved by our experiment. Furthermore, comparison of our data with existing thermopower data [4] shows excellent quantitative agreement *with units*, therefore providing us great confidence in the results presented here. We would like to note that the resistivity of the gold contacts is also negligible. Even if the wiring resonance due to parasitic capacitance at the contact level could contribute to the specific heat, the results of the control experiment discussed in the next point demonstrate the contrary. This is no coincidence as we are indeed probing the specific heat of 2DEG only.
- (d) We have performed a control experiment previously [1,2] in the same Corbino sample, more precisely in the integer quantum Hall (IQHE) in relatively high filling factors. The results were published in the supplementary materials of Ref. [1]. In the light of this comment, we have expanded the SM section associated with this manuscript and we have added these data as well as a brief discussion.

[1] B. A. Schmidt, K. Bennaceur, S. Gaucher, G. Gervais, L. N. Pfeiffer, and K. W. West, Phys. Rev. B 95, 201306 (2017).

[2] B. A. Schmidt, Ph.D. Thesis, McGill University (2019).

[3] F. Schulze-Wischeler, U. Zeitler, C. v. Zobeltitz, F. Hohls, D. Reuter, A. D. Wieck, H. Frahm, and R. J. Haug, Phys. Rev. B 76, 153311 (2007).

[4] W. E. Chickering, J. P. Eisenstein, L. N. Pfeiffer, and K. W. West, Phys. Rev. B 87, 075302 (2013).

2. We are deeply concerned about the entropy analysis.

First, at ultra-low temperatures below 100 mK, any heat capacity measurement unavoidably includes the background, and the unreliability of the thermometer also causes the deviation from the “real” data. Integral from heat capacity to get entropy is controversial because all the errors from heat capacity will accumulate. Heat capacity is more difficult to measure at lower temperatures, and C/T gives more weight for the data with larger uncertainty. We think the authors should provide further evidence in this regard, especially the quantitative evidence.

Second, getting the entropy from the specific heat is dangerous for another reason. In any realistic measurement, people can measure heat capacity to only a lowest limit (let's say ~20 mK in this manuscript). One can only calculate the entropy from 20 mK but there is no apparent way to guarantee C/T below 20 mK is unimportant, which change the shape of the entropy curve after integral.

Author's response: We hope we understood well this comment. In this manuscript, no numerical integration was performed in terms of entropy because we presented the C/T ratio versus T whose area under the curves (or integral) graphically provides the entropy. Here, both C and T were measured separately, as is the norm in other fields such as in superconductivity or superfluidity research. Our goal was to graphically show that there is *simply too much area under the curve for the effective mass to be on the order of m_e* . It is true that we performed an integration or a calculation of the area under the curve up to 200 mK, and this was to check for consistency with the value estimated from the limiting value of the FL. We agree with the referees that the errors grow here from "low" to "high" T , although in the thermopower this could be better estimated. In the light of these comments, we have modified the text and we have re-emphasized our main message which is that there is simply too much area under the curve for the CF effective mass to be closed to m_e , *unless* p_{CF} differs greatly than in the 1st LL. We hope that our revised manuscript addresses all these concerns, and we thank the reviewers for pointing this out.

3. Minor issues:

The authors measured the specific heat between 20mK and 80mK and extended the trend to 200 mK. The trend, i.e. the guide-to-the-eye curve, is very critical in the estimation of the effective mass. We recommend that the authors describe in more details how they decided the trend. Can the authors explain why the impurity scattering parameter p_{CF} is 0 in their sample? As the authors stated in the main text, $p_{CF}=1$ would reduce the value by a half, which could make the effect mass value at $\nu=5/2$ much closer to the original value.

Author's response: We thank the reviewers for pointing this out. The curve describing the trend (guide-to-the-eye) is derived in [1] (p.88) and described in [2]. We modelled the $5/2$ density of states as two flat regions separated with a gap Δ and placed the Fermi level halfway between the two levels defined by the gap. Then, based on this density of states, we derived the temperature dependence of the specific heat as shown in eq.2 of [2]. The complete equation, including the fitting procedure was provided in [2], but we stress this is merely a guide-to-the-eye, and we stress that our conclusions are based on both the specific heat and thermopower data.

To address the second issue, while setting p_{CF} to zero seems arbitrary, this was based on the theoretical work of N.R Cooper [3], in which a value of 0.13 was obtained for this parameter. What we believe the referees propose, is that we cannot rule out a small (near zero) p_{CF} parameter in the first Landau level and closed to 1 in the SLL. In the extreme case of $p_{CF}=1$, we note however that the CF effective mass reported here would still be larger than m_e , but not by much. To address this, we have expanded our discussion of the p_{CF} potential scenarios in the main text. We thank the referee for pointing this out.

[1] B. A. Schmidt, Ph.D. Thesis, McGill University (2019).

[2] B. A. Schmidt, K. Bennaceur, S. Gaucher, G. Gervais, L. N. Pfeiffer, and K. W. West, Phys. Rev. B 95, 201306 (2017).

[3] N. R. Cooper, B. I. Halperin, and I. M. Ruzin, Phys. Rev. B 55, 2344 (1997).

Reviewer #2 (Remarks to the Author):

The manuscript from M. Petrescu et al. reports on the specific heat measurement of the effective mass of composite fermions (CF) in the composite Fermi liquid at the filling factor $\nu = 5/2$. The result is about two to four times the bare electron mass, which is significantly larger than the corresponding value in the half-filled lowest Landau level (LLL).

In my opinion, the experimental procedures and analysis have been clearly explained in the manuscript. The study is also crucial in the understanding of CF theory by highlighting the possible difference between CF in the half-filled LLL and half-filled higher LLs. Therefore, I would recommend a publication of the manuscript in Nature Communications if the authors can address the following comments and suggestions:

Author's response: We thank the reviewer for the positive outlook on our work and for supporting publication in Nature communications.

1. The effective mass was actually estimated by extrapolating the results in the $\nu = 5/2$ FQH state to the high-temperature regime in which the composite Fermi liquid is believed to form. While this assumption seems to be reasonable and further justified by the thermopower data, is there any more direct evidence for this claim? For example, any estimate of the gap $\Delta_{5/2}$ for the $\nu = 5/2$ FQH state in the sample? Is the temperature 200 mK large compared to $\Delta_{5/2}$?

Author's response: We thank the reviewer for asking these questions and we have modified the manuscript accordingly to improve the clarity and to add certain points (see list of changes and added text in red in the revised manuscript). Regarding the energy gap, indeed, we have measured the $\Delta_{5/2}$ energy gap obtained using the standard and accepted methodology with an Arrhenius plot. These measurements are now provided in the supplementary material. Based on this analysis, we obtained a gap of ~ 170 mK which is smaller than 200 mK, the temperature at which we'd expect to crossover towards a CF Fermi liquid. We caution, as is well known in the community, that the energy gaps of relatively "weak" FQH states depend greatly on the cooldown preparation and LED illumination condition of the 2DEG. For instance, and as is now shown in the SM, we can measure in the same Corbino sample energy gaps that differs significantly, however we have found that our specific heat data did not differ much, especially for our measurements above 50 mK or so.

Regarding the second point, we thank the reviewer for providing a great example to further justify the Fermi liquid picture. At temperatures above the many-body energy gap $\Delta_{5/2}$, a Fermi phase of spin-polarized composite fermions (CFs) is believed to occur and the formation of a Fermi Sea

at $\nu = 5/2$ has been confirmed with surface acoustic waves [1] and more recently in a geometric resonance experiment [2]. Taken altogether, it is our view that all data points towards the formation of a CF Fermi liquid phase at temperature above the FQH liquid phase, albeit much of it remains to be understood. This is in large part due to the simple lack of thermodynamic data in the second Landau level in both Fermi liquid (FL) and FQH phases. We believe this is important because at filling fraction $5/2$, both phases can be accessed experimentally in a range of temperatures that is sufficiently low, thanks to the energy gap of the $5/2$ FQH that is not in the Kelvin range.

[1] R. L. Willett, K. W. West, and L. N. Pfeiffer, *Phys. Rev. Lett.* **88**, 066801 (2002).

[2] Md. Shafayat Hossain, Meng K. Ma, M. A. Mueed, L. N. Pfeiffer, K. W. West, K. W. Baldwin, and M. Shayegan, *Phys. Rev. Lett.* **120**, 256601 (2018).

2. I think the C/T curves in Figs. 2 and 3 for $\nu^* = 0$ show a little peak at $T \approx 70$ mK and drop at a higher temperature. This trend looks similar to what happens in the BCS transition. It is hard to rule out this possibility without more data points in higher temperature (beyond 80 mK). Also, the temperature of the apparent peak is close to the transition temperature $T_c \approx 60$ mK predicted from the third law of thermodynamics argument in the manuscript. Can the authors clarify or comment on that?

Author's response: We thank the reviewer for this observation. While it is true that a small peak seems to appear around 70 mK and it would be interesting to investigate the possible links to a BCS transition, the uncertainty in this region is quite large and makes it difficult to draw conclusions about the trend of the specific heat. This has been clarified in the revised manuscript (see text marked in red).

3. In the first paragraph, it is mentioned that the $5/2$ FQHE may be described by a Pfaffian state or its particle-hole symmetric form. Meanwhile, recent numerical results suggested that the $5/2$ FQH state may be described by the anti-Pfaffian state (particle-hole conjugate of the Pfaffian state). Since the nature of this FQH state is still under debate, it is better to mention and include the references on the anti-Pfaffian state as well.

Author's response: We thank the reviewer for this thoughtful suggestion, and we apologize since we should have included them to offer better context. These references [1-2] below have now been included.

[1] M. Levin, B. I. Halperin, and B. Rosenow, *Phys. Rev. Lett.* **99**, 236806 (2007).

[2] S.-S. Lee, S. Ryu, C. Nayak, and M. P. A. Fisher, *Phys. Rev. Lett.* **99**, 236807 (2007).

4. To the best of my knowledge, the derivation of Eq. (1) in the original work by Cooper, Halperin, and Ruzin assumed that the composite fermions have a parabolic dispersion. There is a recent debate on whether the composite fermions are actually Dirac-like or not. Does it matter in the present work? I suggest the authors to clarify the assumption in Eq. (1).

Author's response: This is an excellent point, and we are aware of the current debate regarding whether there is a Dirac-like dispersion at $\nu = 5/2$ or not. To address this, we have modified the manuscript to clearly state the assumptions in Eq.1.

Reviewer #3 (Remarks to the Author):

Reviewer #4 (Remarks to the Author):

Under a high perpendicular magnetic field at very low temperatures, a two-dimensional electron system may enter the fractional quantum Hall state at the Landau level filling factor $\nu=5/2$. Quasiparticles at the $5/2$ state may obey non-Abelian statistics, and thus they have been intensively studied due to the fundamental interest in strongly correlated electron systems and the possible applications toward topological quantum computation. This manuscript reports on the properties of the electron system near $\nu = 5/2$ at high temperatures, where the system does not enter the $5/2$ -fractional quantum Hall state. It is known that, at such high temperatures, the electron system is in a Fermi liquid phase of spin-polarized composite fermions. The authors of the present manuscript claim that, while the effective mass of the composite fermions in the Fermi liquid phase near $\nu = 1/2$ and $3/2$ has been reported in the previous papers, the present manuscript report is the first to provide the experimental estimation of the effective mass near $\nu = 5/2$.

I do not recommend publishing this manuscript in Nature Communications. The reasons are as follows.

Author's response: We appreciate the in-depth analysis of our manuscript and the insightful comments made by the referee. We understand the reviewer's concerns and we will address them all below.

1. This manuscript reports a large effective mass near $\nu = 5/2$ but fails to explain why this is an important finding. Even if the authors are correct in claiming that this is the first observation of the effective mass at $\nu = 5/2$, I cannot accept the result as impactful enough for general readers of Nature Communications without sufficient evidence that the observation is scientifically important.

Author's response: We believe that understanding the physics in the second Landau level is of paramount importance, and this is extremely timely given the recent advance in materials' growth and quality made by our collaborator Dr. Pfeiffer who can now grow GaAs/AlGaAs with electron

mobility exceeding $50 \text{ million cm}^2/(\text{V s})$. We also believe that understanding the CF liquid phase at $5/2$ is extremely important, given the current debate on the exact topological order and nature of this unusual FQH state. Finally, we note in passing that if we opted for a more general audience, we believe that our findings and methods will prove useful for a wide variety of semiconductor experimentalists given that our technique can in principle be applied in other semiconductors at sufficiently low temperatures.

2. There needs to be an explanation of the mechanism why the effective mass is greater at $\nu=5/2$.

Author's response: While we acknowledge that an explanation of the mechanism behind the seemingly larger effective mass observed at $\nu = 5/2$ filling fraction would be valuable, we believe this is beyond the scope of this work. While there has been numerous experimental works performed by a variety of groups in the condensed phase the $\nu=5/2$ FQH, its topological order is still under debate, and as of today there is a total absence of effective mass measurements in the 2nd LL. We strongly believe that our thermodynamic work and data analysis will trigger theory work that could help understand exactly by which mechanism the $5/2$ FQH do indeed form, and about the exact nature of its putative CF Fermi liquid phase.

3. More discussion needs to be provided for the relationship between the fact that the effective mass is large and that the $\nu=5/2$ non-Abelian fractional quantum Hall effect occurs at lower temperatures.

Author's response: We thank the reviewer for mentioning this. We believe we have presented a fair assessment of our data in a dispassionate manner. In analogy with other strongly interacting condensed matter system, it would be tempting to attribute the larger effective mass observed (within the assumption of Eq.1, see comments) with residual electron-electron interactions not entirely contained by the flux attachment in the CF picture (and as expected by some theory work, leading eventually to a paired state). However, we believe the presentation and discussion of our results is appropriate, and that it will help stimulate solid theory work that could provide a more definite answer on the relationship between effective mass from the putative CF Fermi liquid to that of an even-denominator FQH state.

4. Finally, while the data provided in this manuscript may be sufficient to demonstrate the large effective masses qualitatively, it lacks clarity in representing the difference between the first and second Landau levels and precise quantification.

Author's response: We believe this has been addressed in our manuscript, because we have provided a direct comparison in Fig.4 with previous experimental works performed in the first LL. As mentioned above, one could also speculate that the effective mass found here could be linked to the size of quasiparticles that were numerically found in $5/2$ FQH to be rather large (a few magnetic lengths), [1]. We have also addressed in our manuscript the expected \sqrt{B} dependence that is expected in the first LL, and we believe we have presented a proper and fair current state-of-affair regarding the effective measured in high-mobility 2DEGs hosting the FQHE.

[1] J. Nuebler, V. Umansky, R. Morf, M. Heiblum, K. von Klitzing, and J. Smet, Phys. Rev. B 81, 035316 (2010).

It is premature to immediately judge that this paper interests the general readers because of its flavor of the $5/2$ state in fashion. I respect the data presented here, but that is from a specialized perspective on quantum Hall systems. In summary, this paper should be published in a more specialized journal.

Author's response: We respect the referee's view, but we believe that Nature communications is the proper tribune given its open access to everyone, and that our method could be used to perform thermodynamic measurements in other systems, We would also like to mention that if we tackled the great challenge of measuring the specific heat in the $5/2$ FQH it was because of our desire to understand its normal phase. Comparing with other condensed matter systems such as superfluid ^3He , and other unconventional superconducting systems, we found that unlike these, the normal FL phase at $5/2$ filling fraction was truly lacking thermodynamic data to guide theory. We are also fully aware that unlike superfluid ^3He whose FL parameters are nearly all established, we are of the view that our work will generate more research to truly understand the physics of the even-denominator $5/2$ FQH.

Below I comment on some of my concerns.

5. The authors emphasize the value of the effective mass when assuming $P=0$, which was not fair.

Author's response: We understand the referee's concerns, and we have now added a more complete discussion regarding p_{CF} (see comments above to Referee 1, 3).

6. Determining the effective mass based on the guide-to-the-eye lines (blue lines in Fig. 2, for example) seems non-scientific. Isn't there a better way?

Author's response: The guide-to-the-eye curve is based on a physical model for the specific heat that was previously derived by our group (see p. 88 in [1] and eq. 2 in [2]). The derivation and description of this curve will be included in the supplementary materials (see comments above addressing this).

[1] B. A. Schmidt, Ph.D. Thesis, McGill University (2019), available online at gervaislab.mcgill.ca.

[2] B. A. Schmidt, K. Bennaceur, S. Gaucher, G. Gervais, L. N. Pfeiffer, and K. W. West, Phys. Rev. B 95, 201306 (2017).

7. The insets in Fig. 2 are too small to recognize what is inside.

Author's response: We apologize for this, and we thank the reviewer for pointing out this visual issue. Fig.2 has now been adjusted in the revised manuscript.

8. In the first paragraph on page 3, the authors emphasize the reliability of this study by claiming

that the results of the two methods are in quantitative agreement at relatively low temperatures. However, the data in the high-temperature region is important for effective mass estimates, where the two methods do not agree. Therefore, the authors' claim is not fair.

Author's response: In our analysis, both methods point towards a larger effective mass in the 2 LL at $5/2$ filling fraction. Importantly, the samples used in the thermopower, and specific heat experiments have similar electron mobility and density. In both cases we found that either the limiting behaviour of the C/T , and the area under the curve of C/T versus T , leads to a larger CF effective mass.

9. The authors claim in the caption of Figure 3 that there are no specific heat peaks. However, it is not impossible to see a peak in the data. I observe that some data show the peak-like structure, and others do not. The claim sounds biased.

Author's response: We appreciate your observation about the specific heat peaks but unfortunately the uncertainty in the data in this region does not warrant us to make that claim, *i.e.* enough evidence for a peak-like structure (which could be due to BCS-like process or other type of specific heat anomaly). Technically, the experiment will have to be improved either by us, or another team, for the specific heat to be determined at higher temperatures. Recent advances in materials growth may assist here, but improvement in circuitry, and averaging will have to be made. In this regard, we will gladly share all current technical details, several of which are already available online [1], but also newer ones pertaining to the data present in this work.

[1] B. A. Schmidt, Ph.D. Thesis, McGill University (2019), available online at gervaislab.mcgill.ca.

10. The authors compare their data near $5/2$ with data near $3/2$ and $1/2$ in the previous literature, but for an accurate comparison, the authors should obtain data at $3/2$ and $1/2$ by themselves and compare them with the data at $5/2$.

Author's response: We understand the referee's comment, but we believe we have presented the proper state-of-affair for the CF effective mass in the first LL. We also believe the data presented taken by other teams with carefully designed experiments, when properly analysed, all point towards a CF effective mass roughly of the order m_e in the first LL. Unfortunately, our advance in measuring the specific heat at $5/2$ cannot be implemented at $1/2$ and $3/2$ because of the larger magnetic field required, but most importantly because the temperature dependence of the resistivity (or the conductance in a Corbino) at $1/2$ and $3/2$ filling fraction that is too weak at low temperatures for a specific heat measurement based on time-resolved thermal relaxation time measurements.

Author's response: In closing, we appreciate the comprehensive and insightful reviews of our manuscript, and we thank all the referees for their time. We also hope that our clarifications, and revised manuscript, will address the reviewers' concerns and we thank them in advance for considering our work to be published in the open access journal *Nature Communications*.

List of changes:

A red colour is used in the manuscript to indicate the changes made to the text.

Main text

- 1) Modified two sentences in the introduction.

Modified sentence number 1: After more than three decades since its discovery [1], the $5/2$ fractional quantum Hall effect (FQHE) in the second Landau level (SLL) remains the source of extensive research fuelled by its predicted non-Abelian topological order described by a many-body Pfaffian state [2, 3], **anti-Pfaffian [4, 5]**, or its particle-hole symmetric form [6, 7].

Modified sentence number 2: At temperatures above the many-body energy gap $\Delta_{5/2}$, a Fermi liquid phase of spin-polarized composite fermions (CFs) [11–13] is believed to occur and the formation of a Fermi sea at $\nu = 5/2$ has been confirmed by **surface acoustic waves [14] and geometric resonance [15]** experiments.

- 2) Updated a sentence in the '*CFs Fermi liquid specific heat*' section.

Updated sentence: This model assumes a parabolic dispersion and also that electrons are maximally polarized. While there is currently a debate as to whether or not the $\nu = 5/2$ FQHE could host a Dirac-like dispersion [6], it is unclear whether this would affect its putative CF Fermi liquid phase. In the case of the second assumption, the full polarization of the electrons has been validated experimentally at $\nu = 5/2$ by resistively detected NMR experiments [23,24].

- 3) Figure 2 caption modification and figure update.

Modified sentence: the total entropy from 0 to 200 mK from which the effective mass can also be estimated (see main text).

Figure updated: enlargement of the inset plots in the top panel.

- 4) Updated the title of '**Effective mass estimation from C/T and entropy considerations**' section.

- 5) Figure 3 caption modification to avoid any confusions.

Modified a sentence: a large specific heat peak decrease or jump.

- 6) Modified a sentence at the end of the section named '*Effective mass estimation from C/T and entropy considerations*'.

Modified sentence: This being said, we stress that there is simply too much area under the curve of C/T versus T for the CF effective mass to be equal to the bare electron mass (red hatched area). This is true assuming a scattering parameter p_{CF} that is similar to that calculated in the first Landau level at half fillings (see discussion below).

- 7) Modified a sentence in the section '*Considerations in the event of a specific heat anomaly and thermodynamic transition*'.

Modified sentence: we do not observe a clear specific heat anomaly whose decrease would have had to reach the red dashed line in order for $m_{CF} = m_e$.

- 8) Updated Figure 4.

Updated the references in the legend of the plot to match the new order (with the addition of two new references in the introduction, the numbering has been updated).

- 9) Modified a sentence in the section '*Comparison with the first Landau level*'.

Modified sentence: However, we cannot rule out the possibility for p_{CF} to be close to unity in the second LL which, if it were the case, would bring the CF effective masses closer in values. New theory work is certainly required here to clarify the exact role played by p_{CF} in the second Landau level.

- 10) Included data availability statement '**DATA AVAILABILITY**'.

- 11) Included code availability statement '**CODE AVAILABILITY**'.

- 12) Added a sentence in the '*Acknowledgements*'.

Added sentence: F. Boivin for assistance during the preparation of the revised manuscript.

Reference section

- 1) Added new references [4] (Levin et. al. 2007) and [5] (Lee et. al. 2007) for completeness.

Supplementary Material

The supplementary material has been greatly enlarged (and reorganized – *i.e.* the order of sections has been changed) to include:

- 1) Updated figure S1 (sample resistance value for both circuits).
- 2) Modified a sentence and a figure in section – '*Conductance Measurement*'.

Updated existing figure - Fig. S1 (sample resistance value for both circuits).

Modified sentence: The main objective of the latter is to perform time-resolved measurements of the thermal time constant, τ .

- 3) New section – '*Gap Extraction from Arrhenius Fitted Conductance*' and a new figure - Fig. S2.
- 4) New section – '*Thermal Relaxation - Time Constant*' and a new figure - Fig. S4.
- 5) New section – '*Thermal Conductance*' and a new figure - Fig. S5.
- 6) New section – '*Specific Heat in Higher Landau Levels*' and a new figure - Fig. S7.
- 7) Modified sentences in section – '*Uncertainty/Error*'.

Modified sentence 1: A fully detailed overview can be found in Ref. [30, 31].

Modified sentence 2: The thermopower entropy data was determined using a digitized version of the data of Ref. [26], with uncertainty determined by visual inspection of the local noise, and overall background fluctuations.

Modified sentence 3: In any case, all methods that we considered to estimate the effective mass led to m^*/m_e being considerably larger than one, and roughly two to three times larger than in the first Landau level at 1/2 filling factor, provided that the impurity scattering parameter p_{CF} does not greatly differ than in the first Landau level (see main text).

- 8) New section – '*Reproducibility*' and a new figure - Fig. S8.

REVIEWERS' COMMENTS

Reviewer #1 (Remarks to the Author):

The authors have addressed all the comments from my previous referee report.

Reviewer #2 (Remarks to the Author):

After reading the revised manuscript and the supplementary materials, I think they have already addressed my comments and suggestions in the previous report. The authors also replied to some of the concerns by other referees by pointing them to suitable literature. Hence, I will recommend a publication of the manuscript in Nature Communications.

Reviewer #3 (Remarks to the Author):

Reviewer #4 (Remarks to the Author):

I have read the comments from all the reviewers, responses from the authors, and the revised manuscript. While I still think this manuscript would be better suited for a more specialized journal, I acknowledge that the authors have addressed the reviewers' questions and comments and improved the manuscript. It is good that the revised supplement includes a detailed description of the analysis used in this manuscript. My main claim for the revised manuscript is as follows.

As stated in my initial reviewer's report, I understand the significance of the effective composite-fermion mass in the second Landau level for quantum Hall effect experts. However, this manuscript needs to adequately explain why the current findings are important in a broader research context. While I appreciate the authors' attempt to present a fair assessment, giving an impartial data evaluation and avoiding discussions are distinct. If this manuscript is to be published in Nature Communications, I suggest revising it for the following points.

1. The abstract and the introduction state that the effective mass at $5/2$ is yet to be measured. However, the fact that it has not been measured is insufficient to convince general readers that effective mass is essential. The authors wrote, "We strongly believe that our thermodynamic work and data analysis will trigger theory work that could help understand exactly by which mechanism the $5/2$ FQH do indeed form, and about the exact nature of its putative CF Fermi liquid phase" in the response letter. Following this response, I consider that the authors should explain how problematic the lack of effective mass information at $5/2$ in the research field and mention that the present results solved this problem.

2. In their response letter, the authors point out that the observed effective mass values may be related to residual interactions. This point appears in the conclusion. However, such information should be presented much earlier to explain why this experiment is necessary. They also point out that thermodynamic data are lacking for the $5/2$ FQH system, unlike other condensed matter systems. This point is also worth mentioning in the main text. I am not arguing that conclusions should be drawn on these points. I argue that the authors should explain why and how this manuscript is essential for condensed matter physics.

Reviewer #1 (Remarks to the Author):

The authors have addressed all the comments from my previous referee report.

Author's response: We thank both referee 1 and 3 for their joint and thorough review of our manuscript, for their positive comments, and for their appreciation regarding our advance in measuring the specific heat of a 2DEG at very low temperatures.

Reviewer #2 (Remarks to the Author):

After reading the revised manuscript and the supplementary materials, I think they have already addressed my comments and suggestions in the previous report. The authors also replied to some of the concerns by other referees by pointing them to suitable literature. Hence, I will recommend a publication of the manuscript in Nature Communications

Author's response: We thank the reviewer for the positive outlook on our work and for supporting publication in Nature communications.

Reviewer #3 (Remarks to the Author):

Reviewer #4 (Remarks to the Author):

I have read the comments from all the reviewers, responses from the authors, and the revised manuscript. While I still think this manuscript would be better suited for a more specialized journal, I acknowledge that the authors have addressed the reviewers' questions and comments and improved the manuscript. It is good that the revised supplement includes a detailed description of the analysis used in this manuscript. My main claim for the revised manuscript is as follows.

As stated in my initial reviewer's report, I understand the significance of the effective composite-fermion mass in the second Landau level for quantum Hall effect experts. However, this manuscript needs to adequately explain why the current findings are important in a broader research context. While I appreciate the authors' attempt to present a fair assessment, giving an impartial data evaluation and avoiding discussions are distinct. If this manuscript is to be published in Nature Communications, I suggest revising it for the following points.

Author's response: We thank the reviewer for the thoughtful comments and suggestions.

1. The abstract and the introduction state that the effective mass at $5/2$ is yet to be measured. However, the fact that it has not been measured is insufficient to convince general readers that effective mass is essential. The authors wrote, “We strongly believe that our thermodynamic work and data analysis will trigger theory work that could help understand exactly by which mechanism the $5/2$ FQH do indeed form, and about the exact nature of its putative CF Fermi liquid phase” in the response letter. Following this response, I consider that the authors should explain how problematic the lack of effective mass information at $5/2$ in the research field and mention that the present results solved this problem.

Author’s response: We thank the reviewer for this comment, and to address this we have now included a paragraph in the introduction. We also thank the reviewer for overall positive view regarding our work and the thorough and thoughtful review.

2. In their response letter, the authors point out that the observed effective mass values may be related to residual interactions. This point appears in the conclusion. However, such information should be presented much earlier to explain why this experiment is necessary. They also point out that thermodynamic data are lacking for the $5/2$ FQH system, unlike other condensed matter systems. This point is also worth mentioning in the main text. I am not arguing that conclusions should be drawn on these points. I argue that the authors should explain why and how this manuscript is essential for condensed matter physics.

Author’s response: We fully agree with the reviewer’s comment and we have now addressed this in the introductory paragraph.

List of changes:

A red colour is used in the manuscript to indicate the changes made to the text.

Main text

- 1) Added a new paragraph in the introduction.

Added paragraph: The concept of an effective mass is ubiquitous in solid state physics and for years it has been used in semiconductors to understand the transport properties of electrons under the influences of a variety of fields. In the case of clean two-dimensional electron gases (2DEGs) described by Fermi liquid theory, the renormalization of the electron mass into an effective mass m^* due to interactions provides important insights into the dynamics of its elementary excitations called quasiparticles. This effective mass can be linked to several thermodynamic quantities that can be experimentally measured, and it provides important guidance for theory work aimed at understanding the many-body electronic states within a set of conditions such as magnetic fields strength, electron densities, etc. In this regard, much progress was made in the past in the first Landau level (FLL), however the opposite cannot be more true in the second Landau level (SLL) where measurements of m^* are entirely absent. This is the object of this work whereby the effective mass was experimentally estimated in the SLL of a clean 2DEG, and was found to be rather large in comparison to its counterpart in the FLL.

- 2) Figure 1 resize (both panels) and updated lower-case characters for the panel labels.
- 3) Figure 2 caption new sentences and figure resize (all panel and insets).

Added sentences: The error bars for the specific heat are showing the statistical errors propagated from the measurements of τ and K , see the Supplementary Information (SI) section 9. The error bars in thermopower data are the uncertainty from the digitized version.

Figure updated: resize and fonts change to comply with the journal's editorial policy.

- 4) Figure 3 caption new sentence and figure resize.

Added sentence: The data and error bars of C/T are the same as in Fig. 2.

Figure updated: resize and fonts change to comply with the journal's editorial policy.

- 5) Figure 4 caption new sentence and figure resize.

Added sentence: The error bars of the first Landau level are the previously reported values results, see Ref. [18, 19, 21].

Figure updated: resize and fonts change to comply with the journal's editorial policy.

- 6) Modified a sentence in the section '*Comparison with the first Landau level*'.

Modified sentence: However, we cannot rule out the possibility for p_{CF} to be close to unity in the second LL which, if it were the case, would bring the CF effective masses closer in values. New theory work is certainly required here to clarify the exact role played by p_{CF} in the second Landau level.

- 7) New sentences were added in the '**Methods**' section.

Added sentences: The contacts were first patented using UV lithography and then fabricated by e-beam deposition of Ge/Ni/Au/Au layers with corresponding 26/54/14/100 nm thickness. In the last step of the fabrication, the contacts were annealed at two different temperatures: first at 370°C and then at 440°C.

Added sentence: The sample was illuminated by a red LED during cool-down until temperature reached 6 K to enhance the 2DEG density and mobility.

Added sentence: The circuits for all measurements are shown in the SI.

Added sentences: Averaging over a million samples of pulse trains and signals was necessary in order to improve the signal to noise ratio and to reduce the overall uncertainty. Furthermore, the parasitic wire resonance peaks were eliminated using the shift and subtract method presented in SI.

Added sentence: Due to the non-linear temperature dependence of the applied power, a minimum amount of 4 data points was kept for the linear fitting procedure of the thermal conductance K outlined in SI.

Added sentences: The thermopower entropy data was data was determined using a digitized version of the data of Ref. [26], with the uncertainty determined by visual inspection of the local noise, and overall background fluctuation. A guide-to-the-eye fit (as showcased in the inset of Fig. 2 top panel) was used to determine the slope at specific temperatures corresponding to thermopower entropy data. This slope corresponds to the ratio of specific heat and temperature which is presented in Fig. 2 and Fig. 3.

- 8) Updated data availability statement '**Data availability**'.

- 9) Renamed sections and updated their order to comply with the journal's editorial policy.

- 10) All the units of measure (*e.g.* mV) are now in italics for consistency and to comply with the journal's editorial policy.

Reference section

- 1) Updated the format of the reference section to include the publications' title and correct last name and first name(s) initials, to comply with the journal's editorial policy.

Supplementary Material

The supplementary material has been greatly enlarged (and reorganized – *i.e.* the order of sections has been changed) to include:

- 1) Updated the title to comply with the journal's editorial policy.
- 2) Updated figure S2 caption.

Added sentence: **The reported error (blue and black error bars) shows the uncertainty in the conductance measurement.**

- 3) Updated figure S6 caption.

Added sentence: **The error bars for the specific heat are showing the statistical errors propagated from the measurements of τ and K , see section 9 of this supplementary information document.**

- 4) Removed Fig. S7 (only referring to our group's previous publication supplementary material section and relevant figure number).

Updated sentences: **A control experiment was carried out in the same Corbino sample in higher Landau levels. The results of the experiment are shown in the Figure S6 of the supplementary material of Ref. *Phys. Rev. B* **95**, 201306 (2017) published by our group.**

- 5) Updated figure S7 (old S8) caption.

Added sentence: **The error was determined as described in section 9 of this supplementary information document.**